# Epistatic interactions between oxytocin- and dopamine-related genes and trust

**Yuna Koyama**[1], **Nobutoshi Nawa**[1], **Manami Ochi**[2,3], **Pamela J. Surkan**[4], **Takeo Fujiwara**[1,2] *

**1** Department of Public Health, Tokyo Medical and Dental University (TMDU), Tokyo, Japan, **2** Department of Social Medicine, National Center for Child Health and Development, Tokyo, Japan, **3** Department of Health and Welfare Services, National Institute of Public Health, Saitama, Japan, **4** Department of International Health, Johns Hopkins Bloomberg School of Public Health, Baltimore, Maryland, United States of America

* fujiwara.hlth@tmd.ac.jp

**Data Availability Statement:** The data analysed in the current study is not publicly available since it is the part of a population-based study conducted by the corresponding author. It is available if approved from the ethics committee of the National Centre

## Abstract

Trust is an essential human trait. Although research suggests that the interplay between oxytocinergic and dopaminergic systems affects trust formation, little research has focused on epistatic (i.e., gene by gene) interaction effects of oxytocin- and dopamine-related genes on trust. Using a sample of 348 adults (114 men), we aimed to investigate the associations between genetic variants in oxytocin- and dopamine-related genes and the general, neighborhood, and institutional trust with consideration of sex differences. Three-way interaction between oxytocin-related gene genotypes, dopamine-related genotypes, and sex was found for the oxytocin receptor gene (*OXTR*)rs1042778 and the Catechol-O-Methyltransferase gene (*COMT*) rs4680 genotypes (p = 0.02) and for *OXTR* rs2254298 and the dopamine D2 receptor gene (*DRD2*) rs1800497 genotypes (p = 0.01). Further sex-stratified analyses revealed that the interaction between *OXTR* rs1042778 and *COMT* rs4680 genotypes was associated with neighborhood trust among men (p = 0.0007). Also, the interaction between *OXTR* rs2254298 and *DRD2* rs1800497 genotypes was associated with institutional trust among men (p = 0.005). Post-hoc analyses found that men with *OXTR* rs1042778 TG/TT and *COMT* rs4680 GG genotypes reported higher neighborhood trust than those with GG + AG/AA (B = 13.49, SE = 4.68, p = 0.02), TG/TT + AG/AA (B = 23.00, SE = 5.99, p = 0.001), and GG + GG (B = 18.53, SE = 5.25, p = 0.003). Similarly, men with *OXTR* rs2254298 AG/AA and *DRD2* rs1800497 CC genotypes showed higher institutional trust than those with AG/AA + TT/TC (B = 15.67, SE = 5.30, p = 0.02). We could not find any interacting associations among women. While we note that our sample size and candidate gene approach have a potential risk of chance findings, our study provides an important foundation toward further exploration of sex-specific epistatic interaction effects of oxytocin- and dopamine-related genes on trust, indicating the importance of both systems in trust formation.

## Introduction

Trust is an essential human trait and is the basis for two distinctive behaviors that are fundamental to society: group-minded conformity and cooperation across broader interdependent

for Child Health and Development upon reasonable request (phone number: +81-3-3416-0181). The data can not be deposit to somewhere, either public repository, within the manuscript, or as supplemental file because current data contains genetic information, and in combination with other sociodemographic background, individual can be indetify. Based on Japanese law to protect personal information (APPI), it is not possible to depisit the data which can be identify, called sensitive information.

**Funding:** This work was supported by the Research Development Grant for Child Health and Development from the National Center for Child Health and Development (24-12). There was no additional external funding received for this study.

**Competing interests:** There are no conflicts of interest to declare.

networks [1]. Group conformity promotes group cohesion and strengthens punitive responses to social norm violators in the face of external threats [1]. It is rooted in in-group trust that is based on knowledge about a particular object, for example, neighbors (i.e., neighborhood trust) and public or private institutions such as police and media (i.e., institutional trust). Cooperation, on the other hand, provides an opportunity for people to share risks and gain novel benefits, at the cost of possible exploitation of the benefits by free-riders and cheaters [1]. Cooperation necessitates trust across diverse groups even in the absence of prior knowledge about the members of these groups [1,2]. Thus, cooperation is rooted in general trust, the general tendency to rely on others [3,4]. General trust, neighborhood trust, and institutional trust are interrelated [5], and have been shown to be heritable traits [6], suggesting the existence of shared genetic predisposition. However, it is not known which gene or genes are involved in the formation of trust.

Oxytocin, a nine amino-acid neuropeptide synthesized primarily in the hypothalamus, has known as a "social hormone" owing to its involvement in social behaviors such as parenting and mating [7,8], as well as trust [9]. The oxytocin receptor gene (*OXTR*) and the cluster of differentiation 38 gene (*CD38*) are two major genes in the oxytocin signaling pathway [10] and have been widely studied for their relationships with social behaviors [11–13]. However, there have been discrepancies reported in the literature to date. For example, in one study *OXTR* rs53576 A allele carriers showed more social connectedness [14] while in another study A allele carriers exhibited lower empathy [15] and lower levels of optimism and self-esteem [16] and GG genotyped individuals showed higher trust [17]. For *CD38*, one study reported that *CD38* rs3796863 AA/AC genotypes were associated with more altruistic behaviors [18], while another reported no association [19]. Nonetheless, possessing the *CD38* rs3796863 C allele has also been associated with more positive social relationships [20]. Another study showed that *CD38* rs3796863 CC risk alleles were related to less parental touch and the lower risk allele was associated with longer durations of parent-infant gaze synchrony and greater parental care [21]. One possible explanation for this discrepancy may lie in sex differences and all of these studies stratified by sex or have examined sex interaction effects. While some studies found different associations between genotypes and social behaviors between men and women [14,17,20], others reported no differences between the sexes [15,18,19].

These current contradictory findings could potentially be explained by the dopamine system. Dopamine is one of the abundant neurotransmitters in the brain and plays a critical role in the regulation of movement, memory function, motivation, reinforcement learning, reward, attention, emotion, and cognitive function [22]. Oxytocin and dopamine receptors are co-expressed in regions of the brain related to social functioning, such as the striatum and prefrontal cortex, and are associated with similar social and affiliative behaviors [23,24]. Thus, some posit that the dopamine system modulates the effect of oxytocin on social behaviors by enhancing the hedonic value of social interactions and by adjusting one's assessment of the risk of engaging in trust behaviors [23]. For example, carrying certain oxytocin (*OXT*) genotypes was associated with reduced D2/D3 dopamine receptor availability that represents dopamine release in the ventral and dorsal striatum in the face of social cues [25]. Also, intranasal administration of oxytocin regardless of genotype enhanced the activation of the ventral tegmental area, the core region of the mesolimbic dopamine system, in response to social cues, reflecting the salience of social stimuli that predict rewards [26]. These findings suggest a significant interplay between the oxytocinergic and dopaminergic systems in humans. Others have also reported interaction effects of *CD38* rs3796863 and the Catechol-O-Methyltransferase gene (*COMT*) rs4680, genes affecting COMT activity and therefore the degradation of dopamine, on human amygdala activation during social stimuli [27]. However, studies using social behaviors or social cognition as outcomes have been conducted only in rats [28,29].

Therefore, the nature of interactions between oxytocin- and dopamine-related genes on social cognition, especially trust, in humans is not known in a non-experimental context.

Given these findings, the purpose of this study was to examine interactions between genotypes of oxytocin-related genes (*OXTR* and *CD38*) and dopamine-related genes (*DRD2* and *COMT*) in relation to the level of trust (i.e., general trust, neighborhood trust, and institutional trust) among community-dwelling adults. Considering the possibility that the social functions of oxytocin [30] and dopamine [31] may differ between women and men as previously reported, the analyses were stratified by sex.

## Materials and methods

### Participants

Study participants were recruited at the three-month infant health check-up of their child or grandchild, thus including mothers, fathers, and their parents (i.e., the grandparents of the infants). Infants were between 3 and 10 months of age at the time of recruitment, which took place from March 1st, 2013 to March 31st, 2014 at their three-month infant health check-up in Tokyo (an urban area) or Okinawa (a suburban area), Japan. A total of 364 individuals from 115 families participated. After excluding participants whom we could not conduct genetic tests and thus lacked genetic information (n = 16), the analytic sample totaled 348. Two hundred thirty-four participants (67.2%) were women (i.e., mothers/grandmothers) and between 21 and 77 years old (median: 48.5), and 114 participants were men (i.e., fathers/grandfathers) and between 22 and 73 years old (median: 39.0). Approximately one-third of the women had completed junior college or vocational school (35.5%) while 58.8% of the men graduated college or more, which are the most common educational level of females and males respectively. Other demographic data are shown in Table 1. The study was approved by the ethics

**Table 1. Sample characteristics.**

|  | Women (n = 234) | | Men (n = 114) | |
|---|---|---|---|---|
| Age (median, min/max values) | 48.5 | 21.0/77.0 | 39.0 | 22.0/73.0 |
| Prefecture of residence (N, %) |  |  |  |  |
| Tokyo | 108 | 46.2 | 58 | 50.9 |
| Okinawa | 126 | 53.8 | 56 | 49.1 |
| Educational attainment (N, %) |  |  |  |  |
| < High school | 66 | 28.2 | 28 | 24.6 |
| Some college | 83 | 35.5 | 18 | 15.8 |
| University/graduate school | 80 | 34.2 | 67 | 58.8 |
| Missing | 5 | 2.1 | 1 | 0.9 |
| Household income (JPY) (N, %) |  |  |  |  |
| < 4 million | 85 | 36.3 | 33 | 28.9 |
| 4–8 million | 74 | 31.6 | 44 | 38.6 |
| 8 million + | 48 | 20.5 | 33 | 28.9 |
| Missing | 27 | 11.5 | 4 | 3.5 |
| General trust (mean, SD) | 63.3 | 21.3 | 62.9 | 21.8 |
| Neighborhood trust (mean, SD) | 59.1 | 21.2 | 56.0 | 22.2 |
| Institutional trust (mean, SD) | 68.2 | 17.6 | 61.7 | 19.7 |

Abbreviations: JPY, Japanese yen.

committee of the National Centre for Child Health and Development (No. 651) and all participants provided written informed consent.

## Genotyping

Deoxyribonucleic acid (DNA) was extracted from 2 ml of passive drool saliva samples collected at the study site using the Oragnere® saliva sampling kit (DNA Genotek Inc., Ottawa, Canada), which can keep DNA at room temperature. Genotyping of the gene variants of interest (*OXTR* rs53576 (within intron 3 of *OXTR*), rs2254298 (within intron 3 of *OXTR*), rs1042778 (within the 3'-untranslated region (3' UTR) of *OXTR*), *CD38* rs3796863 (within intron 7 on chromosome 4p15), *COMT* rs4680 (within chromosome 22q11.21,16), and *DRD2* rs1800497 (within exon 8 of the ankyrin repeat domain, which is one gene downstream of DRD2)) was performed with the TaqMan™ allele-specific amplification method (Life Technologies, Carlsbad, CA), as previously described [32]. The above-mentioned genes were chosen considering the previously reported association with social functioning. *OXTR* rs53576 genotypes were previously identified as predictors of social connectedness [14] and trust [17]. *OXTR* rs2254298 was associated with social cognition [33]. *OXTR* rs1042778 was associated with sociality [34]. *CD38* rs3796863 was associated with communal behaviors [20]. *COMT* rs4680 and *DRD2* rs1800497 were chosen since COMT and DRD2 were shown to have interacted with the oxytocin system in social functions [27,29]. All genotype frequencies were confirmed to be in Hardy-Weinberg equilibrium (S1 Table). The population was divided into four groups according to the genotypes of single nucleotide polymorphisms (SNPs) in genes related to the oxytocin (*OXTR* and *CD38*) and dopamine (*COMT* and *DRD2*) systems. According to a power calculation with the "pwr" package in R [35], 36 to 46 participants in each group were needed to detect a medium to large effect size ($f^2$ = 0.12 to 0.16) with 80% power for an F-test with 4 groups, at the significant level set to 0.001 (= 0.05/ 8 gene combinations, 3 trust types, 2 sex; based on the Bonferroni correction). In order to keep the sample size comparable for each genotype, the sample was divided into 1) a group consisting of homozygotes for the minor allele and heterozygotes for the minor and major alleles, and 2) another group consisting only of homozygotes for the major allele. Since SNP data for *OXTR* rs2254298 and *CD38* rs3796863 were not available from 2 and 3 participants, respectively, sample sizes vary for each analysis on these SNPs.

## Trust

Participant trust (i.e., general, neighborhood, and institutional trust) was assessed based on a self-reported questionnaire. Questionnaires were mailed to participants prior to the visit and completed questionnaires were collected at the study site. Participants responded to 10 trust-related questions, reflecting the level of trust in others in general, in their neighbors, and in government agencies. Scale items were developed based on validated scales for measuring trust [36,37]. All items for each dimension of trust are presented in S2 Table. The score for each item was normalized to 0–1 and summed to obtain general, neighborhood, and institutional trust scores (Cronbach's alpha, general: 0.70; neighborhood: 0.82; institutional: 0.84), which were further normalized to 0–100. The correlations among each dimension of trust were as follows: general and neighborhood trust: 0.45; general and institutional trust: 0.38; neighborhood and institutional trust: 0.32 (all p-values < 0.05). Higher scores indicate higher levels of trust.

## Statistical analysis

In order to detect the epistatic interaction effects, we applied multivariate linear regression with an interaction term between oxytocinergic and dopaminergic genes, as well as the

interaction with sex. The analysis was conducted for men and women, separately where the significant three-way interaction existed. Additionally, an analysis of variance (ANOVA) was performed to examine the mean differences in trust scores across groups characterized by each genotype. ANOVA was followed by Tukey's multiple comparison post hoc tests to identify groups with significantly different levels of trust with the R package "multcomp" [38]. To take into account clustering at the family level and age differences, the sandwich variance estimator was applied, and age was adjusted for all models. All analyses were conducted using R version 4.0.5 [39].

## Results

Interactions between oxytocin- and dopamine-related gene genotypes were explored (S3 Table). We could not find any significant interaction in the general population. However, the interaction between *OXTR* rs1042778, *COMT* rs4680, and sex on neighborhood trust (unstandardized coefficient (B) = 26.38, t = 2.40, p = 0.02), and between *OXTR* rs2254298, *DRD2* rs1800497, and sex on institutional trust (B = -21.04, t = -2.54, p = 0.01) were found. Therefore, we further examined these genotypes by sex.

We found an interaction between *OXTR* rs1042778 and *COMT* rs4680 on neighborhood trust scores among men (B = 28.05, t = 3.38, p = 0.0007), but not among women (B = 1.42, t = 0.18, p = 0.86; Table 2). Men with *OXTR* rs1042778 TG/TT and *COMT* rs4680 GG genotypes showed higher neighborhood trust compared to any other genotyped groups (vs GG + AG/AA: mean difference ± SE = 13.49 ± 4.68, t = 2.88, p = 0.02; vs TG/TT + AG/AA: mean difference ± SE = 23.00 ± 5.99, t = 3.84, p = 0.001; vs GG + GG: mean difference ± SE = 18.53 ± 5.25, t = 3.53, p = 0.003; Fig 1A and S4 Table).

An interaction between *OXTR* rs2254298 and *DRD2* rs1800497 genotypes was found among men (B = -19.46, t = -2.82, p = 0.005), but not among women (B = -0.08, t = -0.02, p = 0.99; Table 3). Men with *OXTR* rs2254298 AG/AA and *DRD2* rs1800497 CC genotype showed higher institutional trust than those with *OXTR* rs2254298 AG/AA and *DRD2* rs1800497 TT/TC (mean difference ± SE = 15.67 ± 5.30, t = 2.96, p = 0.02; Fig 1B and S4 Table).

## Discussion

In this study, we observed interactions between *OXTR* rs1042778 and *COMT* rs4680 related to neighborhood trust, and between *OXTR* rs2254298 and *DRD2* rs1800497 related to institutional trust in men, but not in women. To the best of our knowledge, this is the first study to examine the gene-by-gene interaction of oxytocin- and dopamine-related genes on different levels of trust among humans in a non-experimental setting, with consideration of potential sex differences and variations regarding the type of trust.

**Table 2. Epistatic interaction effects between *OXTR* rs1042778 and *COMT* rs4680 on neighbourhood trust by sex.**

|  | Female | | Male | |
|---|---|---|---|---|
|  | **B** | **P-value** | **B** | **P-value** |
| *OXTR* rs1042778 (TG/TT, ref: GG) | 0.21 | 0.97 | -9.52 | 0.10 |
| *COMT* rs4680 (GG, ref: TG/TT) | 1.09 | 0.72 | -5.04 | 0.31 |
| *OXTR* rs1042778 × *COMT* rs4680 | 1.42 | 0.86 | **28.05** | **0.0007** |

Bold signifies p < 0.05.

All models were adjusted for age.

Standard errors were calculated with a sandwich variance estimator. B stands for unstandardized coefficient.

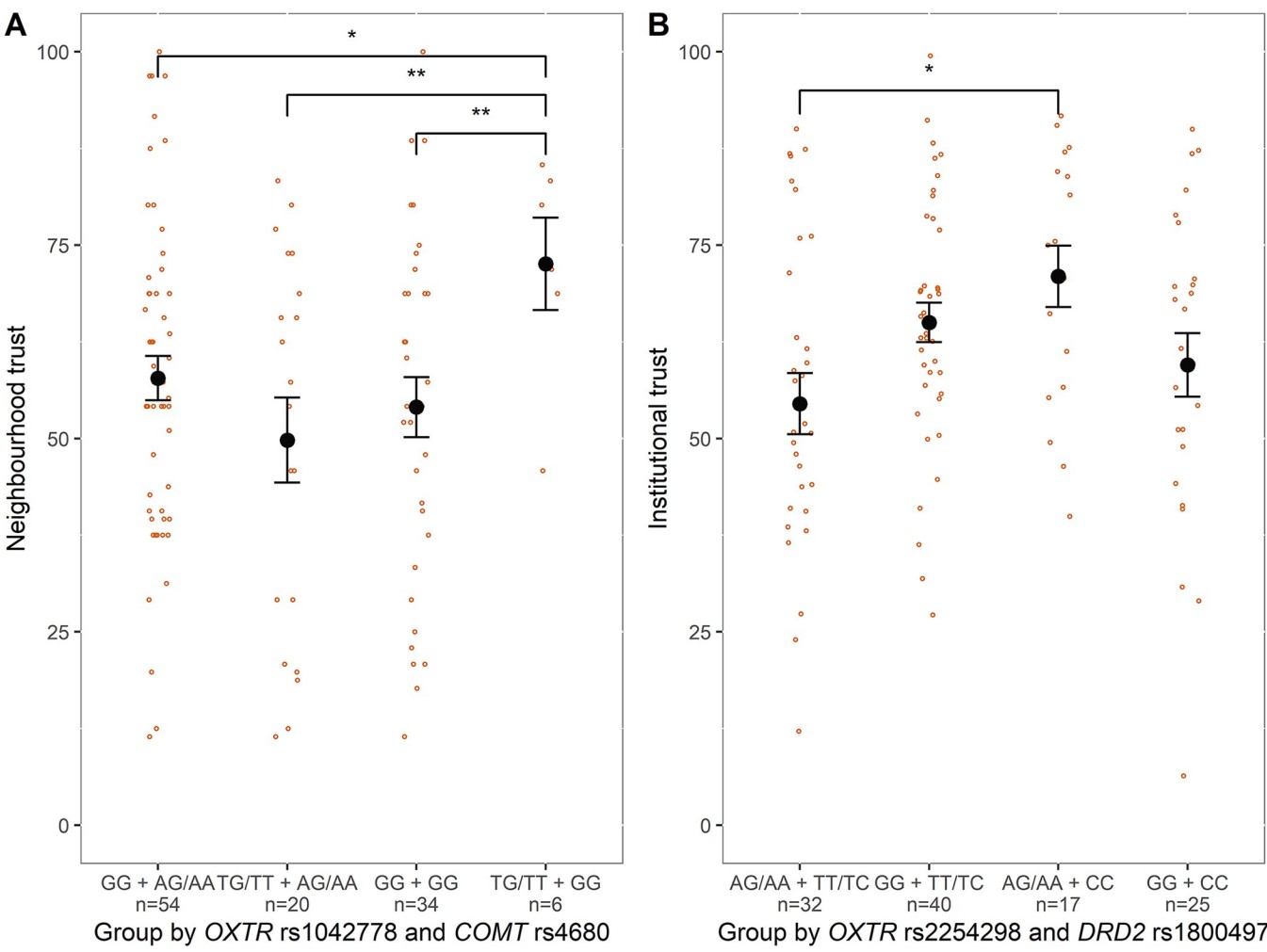

**Fig 1. Mean and distribution of trust scores across genotyped groups among males.** A) Neighbourhood trust scores across male groups genotyped by OXTR rs1042778 and COMT rs4680. B) Institutional trust scores across males genotyped by OXTR rs2254298 and DRD2 rs1800497. Orange and black dots represent individual and mean scores, respectively. Error bars denote standard errors estimated with a sandwich variance estimator. * and ** correspond to a p-value < 0.05 and < 0.01, respectively (adjusted with Tukey's method).

COMT is an enzyme responsible for the methylation of dopamine and regulates dopamine degradation [19,27]. *COMT* rs4680 polymorphisms are associated with enzyme activity, and A allele carriers have demonstrated reduced enzyme activity, leading to reductions in the

**Table 3. Epistatic interaction effects between *OXTR* rs2254298 and *DRD2* rs1800497 on institutional trust by sex.**

|  | Female | | Male | |
|---|---|---|---|---|
|  | **B** | **P-value** | **B** | **P-value** |
| *OXTR* rs2254298 (GG, ref: AG/AA) | 2.78 | 0.37 | **11.66** | **0.01** |
| *DRD2* rs1800497 (GG, ref: AG/AA) | -3.43 | 0.25 | **15.67** | **0.004** |
| *OXTR* rs2254298 × *DRD2* rs1800497 | -0.08 | 0.99 | **-19.46** | **0.005** |

Bold signifies p < 0.05.

All models were adjusted for age.

Standard errors were calculated with a sandwich variance estimator. B stands for unstandardized coefficient.

degradation of dopamine [19,27]. Reductions in dopamine degradation further indicate higher synaptic dopamine availability [19,27], suggestive of higher dopaminergic functioning. In the present study, we found evidence of an interaction between *OXTR* rs1042778 and *COMT* rs4680 on neighborhood trust in men. *OXTR* rs1042778 TG/TT and *COMT* rs4680 GG genotypes were associated with the highest level of neighborhood trust compared to their counterparts. The *OXTR* rs1042778 T allele has been associated with poor sociability [40,41], suggestive of lower levels of trust, albeit without a clear biological role of the polymorphism. Therefore, we had expected that *OXTR* rs1042778 TG/TT genotype carriers would show lower levels of trust, which was not the case in our analysis. Previously reported associations between the *OXTR* rs1042778 T allele with anti-social and aggressive behaviors [42], and lower sociability concerning the general population [34,40,43] might be explained by T allele carriers' higher sensitivity to social stimuli, which was indicated by heightened amygdala reactivity to angry facial expressions [42]. The association between *OXTR* and social attention was also shown in an animal experiment [44], suggesting this as a well-conserved social response across species. Given that we assessed neighborhood trust (i.e., trust toward well-known people), and that the T allele may be linked to higher sensitivity to social environmental stimuli, it is plausible that the association between the T allele and higher neighborhood trust would be observed in Japan, where the degree of social conformity is relatively high [45]. Our results extended the current literature by suggesting that the *OXTR* rs1042778 polymorphism alters levels of trust when combined with GG on *COMT* rs4680, i.e., when dopamine functioning is normal compared with heightened.

*DRD2* encodes the D2 subtype of the dopamine receptor (DRD2). *DRD2* rs1800497, the most extensively studied polymorphism, can be a marker of the number of dopamine receptors in the striatum [46,47]. Specifically, the T allele is associated with a decrease in the number of DRD2 [46,47], and thus a decreased level of dopamine functioning. The current study revealed that men with *OXTR* rs2254298 AG/AA and *DRD2* rs1800497 TT/TC showed lower institutional trust than men with *OXTR* rs225498 AG/AA and *DRD2* rs1800497 CC. Similar to the *OXTR* rs1042778 T allele, the *OXTR* rs2254298 G allele has been tied with poor sociability and, thus lower levels of trust [40,41]. Therefore, *OXTR* rs2254298 AG/AA genotypes would indicate higher levels of trust. Our findings showed social functions of *OXTR* rs2254298 A allele might be stronger in the absence of strong dopamine functions, i.e., with *DRD2* rs1800497 TT/TC genotypes.

Several prior studies have reported oxytocin-dopamine interactions on emotional responses in individuals. A study of European young men showed that men with *CD38* rs3796863 CC (previously identified as a risk genotype) and *COMT* rs4680 AA, and men with *CD38* rs3796863 AA/AC and *COMT* rs4680 GG showed stronger activation of the amygdala, which governs the emotional process, during the presentation of social stimuli than their counterparts carrying *CD38* rs3796863 AA/AC and *COMT* rs4680 AA, and *CD38* rs3796863 CC and *COMT* rs4680 GG, respectively [27]. Another study found that participants with more striatal dopamine transporter availability were more prone to neuroticism only among *OXTR* rs53576 AA carriers [48]. We did not observe exactly the same association. However, the association between risk alleles on oxytocin-related genes and poor social functioning in people with low to standard levels of dopamine functioning is consistent with our findings. Furthermore, this association was attenuated, albeit not reversed in our study sample, among people with high dopamine functioning levels.

These findings on the interaction between oxytocin- and dopamine-related gene genotypes suggest that oxytocin-related genes determine the basis of trust in an individual by altering one's tendency for social affiliation and the ability to process social information [10]. One's ability to social process can be modified by a propensity to engage in risk-taking behaviors [2],

which may be facilitated by levels of dopamine functioning. Some of the social functions governed by dopamine are attributable to reward-prediction errors that bias behavioral selection toward stimuli that predict a reward [49]. Thus, lower levels of dopamine functioning may strengthen the association between the oxytocin risk allele and lower levels of trust since people with lower levels of dopamine functioning tend not to take risks and do not view social interactions as rewarding. In contrast, higher levels of dopamine functioning may weaken the association between the oxytocin risk allele and lower levels of trust because people with higher levels of dopamine functioning tend to take risks and trust people in general (as they are motivated by their ability to get more rewards from social interactions).

Prior research suggests that oxytocin [30] and dopamine [31] have sex-specific associations with social behaviors, which may explain the differential associations by sex we observed in the current study. In particular, it has been hypothesized that there is an inverted U-shaped association between brain oxytocin level and social functions that explains sex differences in sensitivity to oxytocin administration, i.e. suggesting that females can enhance social behaviors more sensitively with oxytocin administration, while males require higher doses of oxytocin to change their behaviors [30,50]. Considering sex differences in social functioning from an evolutionary viewpoint, this sex difference in sensitivity seems reasonable, as females are more empathic and sensitive to social signals and choose fewer close social relationships, whereas males prefer aggressive behavior, risk-taking, and more extensive but less intimate relationships [51]. The current focus is on gene variants related to oxytocin and dopamine systems that do not necessarily correspond to the level of exogenously administered and centrally available oxytocin and dopamine and their levels of functioning. Therefore, an alternative explanation would be that existing sex differences in signaling and neural pathways could modify social behaviors. The current study did not uncover any sex-specific mechanisms and we cannot rule out the possibility of chance findings due to the small sample size in each group. Therefore, replication studies with larger sample sizes to examine potential sex differences are needed.

We must be aware of some limitations. First, general trust was assessed with two questions. Although these items were based on prior research [36,37], they may not perfectly capture the concept of general trust with only two items. Further, assessing the act of trust in experimental settings, such as the Trust Game, should be considered [37]. Second, we could not examine the possible epistatic interaction effects on DNA methylation [52], and other epistatic interactions that were not assessed here (e.g., between *COMT* and *DRD2* on emotionality [53]). Thus, future replication studies were warranted. Third, our study sample was exclusively Japanese. Generalizing the current findings needs caution with the consideration of cultural differences in trust and ethnic variations in genes. Fourth, we interpreted gene variants as estimated dopamine levels, i.e., there may be a discrepancy between our estimates of dopamine function deduced from genotypes versus the assessment of actual dopamine levels. However, considering the challenges of measuring dopamine levels in the living human brain, the use of gene variants may be advantageous because they are not subject to fluctuations due to biological responses to internal and external environmental changes (i.e., negative and positive feedback) and allow us to exclude reverse causation (e.g., that higher levels of trust lead to more social interactions, which result in higher peripheral oxytocin levels [54]). The use of human samples is another strength of our research considering that trust in humans and other animals has potentially different meanings and the neural system processes them differently.

## Conclusions

Our study found an interaction between *OXTR* rs1042778 and *COMT* rs4680 on the level of neighborhood trust and an interaction between *OXTR* rs2254298 and *DRD2* rs1800497 on

institutional trust among men. Although we need caution in the interpretation due to the small sample size, our results provided preliminary findings that set the stage to further explore the epistatic interactions between oxytocin- and dopamine-related gene variants on social phenotypes, such as trust, by underscoring the importance of the dopamine system as well as the oxytocin system in the formation of trust. Given the growing significance of understanding how trust influences multidimensional human activities, further research is needed to fully elucidate the mechanisms of trust formation in humans.

## Supporting information

**S1 Table. Frequency and distribution of genotypes of oxytocin- and dopamine-related genes.**
(DOCX)

**S2 Table. Trust questionnaire items.**
(DOCX)

**S3 Table. Epistatic interaction effects between oxytocin- and dopamine-related genotypes on trust.**
(DOCX)

**S4 Table. Trust mean score by genotyped groups.**
(DOCX)

## Acknowledgments

We thank all study participants and their family members who took part in this study.

## Author Contributions

**Conceptualization:** Yuna Koyama, Takeo Fujiwara.

**Data curation:** Manami Ochi, Takeo Fujiwara.

**Formal analysis:** Yuna Koyama.

**Funding acquisition:** Takeo Fujiwara.

**Project administration:** Takeo Fujiwara.

**Supervision:** Takeo Fujiwara.

**Visualization:** Yuna Koyama.

**Writing – original draft:** Yuna Koyama.

**Writing – review & editing:** Nobutoshi Nawa, Manami Ochi, Pamela J. Surkan, Takeo Fujiwara.

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
