## [Decision Letter · Decision Letter 0]

1 Apr 2024

PONE-D-23-41164Epistatic interactions between oxytocin- and dopamine-related genes and trustPLOS ONE

Dear Dr. Fujiwara,

Thank you for submitting your manuscript to PLOS ONE. After careful consideration, we feel that it has merit but does not fully meet PLOS ONE’s publication criteria as it currently stands. Therefore, we invite you to submit a revised version of the manuscript that addresses the points raised during the review process.

Please specifically address the concerns raised by reviewers considering the issue of the small sample size and validity of the obtained results. ==============================

We look forward to receiving your revised manuscript.

Kind regards,

Alexandra Kavushansky, PhD

Academic Editor

PLOS ONE

Journal Requirements:

“This work was supported by the Research Development Grant for Child Health and Development from the National Center for Child Health and Development (24-12).”

3. In this instance it seems there may be acceptable restrictions in place that prevent the public sharing of your minimal data. However, in line with our goal of ensuring long-term data availability to all interested researchers, PLOS’ Data Policy states that authors cannot be the sole named individuals responsible for ensuring data access (http://journals.plos.org/plosone/s/data-availability#loc-acceptable-data-sharing-methods).

“This work was supported by the Research Development Grant for Child Health and Development from the National Center for Child Health and Development (24-12).”

“This work was supported by the Research Development Grant for Child Health and Development from the National Center for Child Health and Development (24-12).”

Reviewers' comments:

Reviewer's Responses to Questions

**Comments to the Author**

1. Is the manuscript technically sound, and do the data support the conclusions?

Reviewer #1: No

Reviewer #2: Partly

Reviewer #3: No

2. Has the statistical analysis been performed appropriately and rigorously? 

Reviewer #1: I Don't Know

Reviewer #2: No

Reviewer #3: Yes

3. Have the authors made all data underlying the findings in their manuscript fully available?

Reviewer #1: No

Reviewer #2: No

Reviewer #3: No

4. Is the manuscript presented in an intelligible fashion and written in standard English?

Reviewer #1: Yes

Reviewer #2: Yes

Reviewer #3: Yes

5. Review Comments to the Author

Reviewer #1: Please see the attached file.

Reviewer #2: Overall a nice paper but some clarification needed in places. Supp tables are overviews rather than data so this may needed added in to satisfy point 3 about data underlying findings.

Line 23: OXTR and COMT not defined in abstract

Line 24: DRD4 not defined in abstract

Line 46: Citation for sentence ending “… free-riders and cheaters”.

Line 54: CD not defined

Line 63: consider adding “as” to make clearer. “… sex differences as all of these studies….”

Line 80: COMT not defined

Line 82: clarification of use of word “endpoints”

Line 95: confusing first line of method. Once whole section is read meaning is clear, but it adds confusion with the first line wording as is. Perhaps “Study participants were recruited at the three-month infant health check-up of their child or grandchild”.

Line 99: is it standard for genetic information to be known about citizens in Japan? Maybe add a sentence here as this is unusual in other parts of the world.

Lines 100-102: add mean age of participants with the range

Line 102 – 103: for females junior college and vocational reported and for males graduated college reported. Consider reporting same education level for M&F e.g. how many females graduated college so that it is comparable. Or emphasises that these were the most common categories for both sexes.

Line 109: How were the saliva samples treated and stored?

Line 110: specify the type of variant (SNPs but also what type e.g. upstream, 5’ UTR etc).

Line 112: were PCR protocols optimised or were they identical to reference 30? Gel % and stain type, primers all the same?

Line 144: stats – was relatedness between individuals accounted for? i.e. if there was a mother and grandmother that were themselves mother and daughter. If participants were not related specify this in participant info section. If individuals were related, consider a MCMC-based Bayesian GLMM with priors to reduce chance of type 1 errors. If this stats test is not possible comment on how relatedness was accounted for in the analysis.

Line 145: unclear why multivariate linear regression was used in addition to ANOVA when the variables used were categorical (genes and sex) and therefore more suited to the ANOVA. If these are not categorical perhaps a data prep section would help with clarification here.

Line 151: reference needed for package

Line 158-159: no test statistic or dfs reported with results.

Line 162: which test does the B value come from. As far as I am aware you get t from LR and f from ANOVA. If another test was conducted this needs reported in the stats section. If B is something other than the test statistic this needs explained here. Same in Tables 2 and 3.

Line 162: Consider “…among women (B=1.42, p = 0.86; Table 2).” Rather than having brackets back to back. Same on line 166 and 172.

Line 164-165: report df for genotypes and for M&F for the genotypes on line 169-172.

Line 190 (OXTR discussion): consider also OXTR has been shown to be linked with social attention in rhesus macaques (Howarth et al., 2023; 10.1371/journal.pone.0288108) suggesting that this is a well conserved social response across species.

Line 219: A allele carries for OXTR rs53576 were also had lower level of optimism and self-esteem (Saphire-Bernstein et al., 2011; https://doi.org/10.1073/pnas.1113137108) and have poor social recognition (Skuse et al., 2014; 10.1073/pnas.1302985111).

Line 255: how might they not capture the concept of general trust?

Line 265: why is using humans a strength?

Figure 1 is blurry but downloadable version is fine

Reviewer #3: The study used a candidate gene family study approach to select variants in biological systems that have been strongly implicated in trust and test for an epistatic interaction to understand genetic contributions to self-report trust measures, stratified by sex. The justification for the biological systems selected is sound and well-explained. However, unfortunately the changing conventions and standards of evidence in the field of behavior genetics draw immediate concerns about the strength of this study.

Candidate gene association studies have received intense scrutiny for issues with replication, with genetics journals resistant to publishing these types of studies at all for the past dozen years (Hewitt, 2012). More recent meta-analytic evidence revisiting the topic of candidate gene approaches vs. genome wide association study (GWAS) also urges researchers to discontinue the use of candidate gene approaches (van de Weijer, 2022). For even longer, the best practice standard for candidate gene studies has insisted on sample sizes of at least 1,000 participants (Duncan & Keller, 2011). Although family studies with a candidate gene approach may have value, current scientific attitudes advocating for these types of approach say that sample sizes in the thousands are expected, and the evidence is more valuable if not relying solely on a candidate gene approach but instead working in concert with GWAS evidence (Friedman, et al., 2021).

Due to the interaction terms in this study and running analyses separately by gender, the already small sample size of 364 individuals has further reduced statistical power, leading to concern that the significant findings are false positives, as appears to be the case for most findings from studies of these types based on failures to replicate (Border et al., 2019, Hatoum et al., 2020; Johnson, et al., 2017; Farrell et al., 2015; Culverhouse, et al., 2018; Duncan & Keller, 2011; Duncan et al., 2019).

Based on the fundamental failure of this study to meet the current standards of behavior genetics research, I unfortunately cannot recommend publication. For serious consideration of this evidence, I would need to at least see an independent replication with a sample size in the thousands, but most geneticists would still be hesitant to give this evidence much confidence without supporting GWAS evidence. I would recommend the authors read the paper by Friedman et al., 2021 (https://doi.org/10.1016/j.tics.2021.06.007), if they have not already, and consider revising their approach to these research questions. I wish the researchers the best luck; conducting research in such a rapidly changing field undergoing major paradigm shifts is challenging but based on the researchers’ clear expertise in the biological systems of interest, I am confident they will produce other valuable work in the future.

6. PLOS authors have the option to publish the peer review history of their article (what does this mean?). If published, this will include your full peer review and any attached files.

Reviewer #1: No

Reviewer #2: **Yes: **Emmeline Howarth

Reviewer #3: No

---

## [Author Response · Author response to Decision Letter 0]

25 Jun 2024

Rebuttal letter

Reviewer #1

The manuscript by Koyama et al. attempts to establish a link between the oxytocinergic and dopaminergic systems, hypothesizing an epistatic relationship between the corresponding genes. These gene sets play key roles in shaping behaviours of key societal significance. The study’s findings highlight the presence of an interaction between OXTRrs1042778 and COMTrs4680 concerning neighbourhood trust. An interaction was also observed between OXTRrs2254298 and DRD2rs1800497 impacting institutional trust, but exclusively among males. While these results, albeit limited, may be of interest to a specialized audience, the manuscript is not without its limitations. 

Response: 

Thank you so much for acknowledging the potential interests of our findings. We tried to address the points you raised in the following manners.

#1. Firstly, the authors devote scant attention to the literature on CD38 and its role in confidence, a subject extensively discussed by biologists and medical groups working in different fields. Moreover, references to most of the pivotal papers on the subject are conspicuously absent.

Response:

Thank you for your point. We searched the literature again to cover the crucial articles about CD38 and social relationships in the introduction.

“Another study showed that CD38 rs3796863 CC risk alleles were related to less parental touch and the lower risk allele was associated with longer durations of parent-infant gaze synchrony and greater parental care [21].” (Introduction: paragraph 2)

#2. Another limitation is the failure to analyse the results in relation to the sex of the subjects. 

Response:

Thank you for pointing out the limitation. We conducted formal statistical tests of effect modification by sex (inclusion of interaction terms by sex) and stratified analyses where the interaction was significant with the potential sex differences in mind. However, as stated in the discussion, we could not exclude the potential chance findings due to the small sample size. Given this limitation, we toned down our discussion regarding the sex differences in the conclusion and abstract.

“Although we need caution in the interpretation due to the small sample size, our results provided preliminary findings that set the stage to further explore the epistatic interactions between oxytocin- and dopamine-related gene variants on social phenotypes, such as trust, by underscoring the importance of the dopamine system as well as the oxytocin system in the formation of trust.” (Conclusions)

“While we note that our sample size and candidate gene approach have a potential risk of chance findings, our study provides an important foundation toward further exploration of sex-specific epistatic interaction effects of oxytocin- and dopamine-related genes on trust, indicating the importance of both systems in trust formation.” (Abstract)

#3. A second, and arguably more critical limitation, is the exclusive focus on Japanese individuals in the analysis. While this is not problematic per se and may contribute valuable insights within the context of Japan, it makes the conclusions difficult to extrapolate to societies lacking the peculiar characteristics of Japanese society. These factors hinder the extension of the study’s results to a broader audience. The authors acknowledge additional limitations at the conclusion of an extensive discussion.

Response:

Thank you for raising the important limitation in the current study. Since trust is one of the phenotypes that is strongly affected by social contexts and genes are heavily rooted in ethnic differences, we further acknowledged and added this limitation regarding the generalizability to the discussion.

“Third, our study sample was exclusively Japanese. Generalizing the current findings needs caution with the consideration of cultural differences in trust and ethnic variations in genes.” (Discussion: paragraph 7)

Reviewer #2: 

#1. Overall a nice paper but some clarification needed in places. Supp tables are overviews rather than data so this may needed added in to satisfy point 3 about data underlying findings.

Response: 

Thank you for your kind compliments on our overall manuscript. We revised the manuscript regarding your points as follows.

#2. Line 23: OXTR and COMT not defined in abstract

#3. Line 24: DRD4 not defined in abstract

Response:

Thank you for your point. We added the full names for the first abbreviations in the abstract.

“Three-way interaction between oxytocin-related gene genotypes, dopamine-related genotypes, and sex was found for the oxytocin receptor gene (OXTR)rs1042778 and the Catechol-O-Methyltransferase gene (COMT) rs4680 genotypes (p = 0.02) and for OXTR rs2254298 and the dopamine D2 receptor gene (DRD2) rs1800497 genotypes (p = 0.01).” (Abstract)

#4. Line 46: Citation for sentence ending “… free-riders and cheaters”.

Response:

We added the reference for that sentence.

“Cooperation, on the other hand, provides an opportunity for people to share risks and gain novel benefits, at the cost of possible exploitation of the benefits by free-riders and cheaters [1].” (Introduction: paragraph 1)

#5. Line 54: CD not defined

Response:

We added the full name for CD38.

“The oxytocin receptor gene (OXTR) and the cluster of differentiation 38 gene (CD38) are two major genes in the oxytocin signaling pathway [10] and have been widely studied for their relationships with social behaviors [11–13].” (Introduction: paragraph 2)

#6. Line 63: consider adding “as” to make clearer. “… sex differences as all of these studies….”

Response:

Thank you for your suggestion. We connected two sentences so that our implied contents would be better conveyed.

“One possible explanation for this discrepancy may lie in sex differences and all of these studies stratified by sex or have examined sex interaction effects.” (Introduction: paragraph 2)

#7. Line 80: COMT not defined

Response:

We added the full name for the abbreviation.

“Others have also reported interaction effects of CD38 rs3796863 and the Catechol-O-Methyltransferase gene (COMT) rs4680, genes affecting COMT activity and therefore the degradation of dopamine, on human amygdala activation during social stimuli [27].” (Introduction: paragraph 3)

#8. Line 82: clarification of use of word “endpoints”

Response:

In order to clarify the meaning of a sentence, we changed the word from “endpoints” to “outcomes”.

“However, studies using social behaviors or social cognition as outcomes have been conducted only in rats [28,29].” (Introduction: paragraph 3)

#9. Line 95: confusing first line of method. Once whole section is read meaning is clear, but it adds confusion with the first line wording as is. Perhaps “Study participants were recruited at the three-month infant health check-up of their child or grandchild”.

Response:

Thank you for your suggestion and for pointing out the potential confusion. We followed your suggestion and clarified the recruiting process.

“Study participants were recruited at the three-month infant health check-up of their child or grandchild, thus included mothers, fathers, and their parents (i.e., the grandparents of the infants).” (Materials and methods: Participants)

#10. Line 99: is it standard for genetic information to be known about citizens in Japan? Maybe add a sentence here as this is unusual in other parts of the world.

Response:

Thank you for pointing out the potential unclear sentence. It is not standard for Japanese citizens to know genetic information. The genetic information was obtained exclusively among the study participants by our team. We made this point clearer.

“After excluding participants whom we could not conduct genetic tests and thus lacked genetic information (n = 16), the analytic sample totaled 348.” (Materials and methods: Participants)

#11. Lines 100-102: add mean age of participants with the range

Response:

We added the median age of participants in addition to the range of ages with the consideration of skewed distributions.

“Two hundred thirty-four participants (67.2%) were women (i.e., mothers/grandmothers) and between 21 and 77 years old (median: 48.5), and 114 participants were men (i.e., fathers/grandfathers) and between 22 and 73 years old (median: 39.0).” (Materials and methods: Participants)

#12. Line 102 – 103: for females junior college and vocational reported and for males graduated college reported. Consider reporting same education level for M&F e.g. how many females graduated college so that it is comparable. Or emphasises that these were the most common categories for both sexes.

Response:

We picked up these categories since they are the most common educational level categories of females and males. We made emphasis on this point.

“Approximately one-third of the women had completed junior college or vocational school (35.5%) while 58.8 % of the men graduated college or more, which are the most common educational level of females and males respectively.” (Materials and methods: Participants)

#13. Line 109: How were the saliva samples treated and stored?

Response:

Saliva samples kept in a saliva sampling kit were stored at room temperature. This is described in Method as follows:

“Deoxyribonucleic acid (DNA) was extracted from 2 ml of passive drool saliva samples collected at the study site using the Oragnere® saliva sampling kit (DNA Genotek Inc., Ottawa, Canada), which can keep DNA at room temperature.” (Materials and methods: Genotyping)

#14. Line 110: specify the type of variant (SNPs but also what type e.g. upstream, 5’ UTR etc).

Response:

We added the explanation of location to the SNPs.

“Genotyping of the gene variants of interest (OXTR rs53576 (within intron 3 of OXTR), rs2254298 (within intron 3 of OXTR), rs1042778 (within the 3’-untranslated region (3’ UTR) of OXTR), CD38 rs3796863 (within intron 7 on chromosome 4p15), COMT rs4680 (within chromosome 22q11.21,16), and DRD2 rs1800497 (within exon 8 of the ankyrin repeat domain, which is one gene downstream of DRD2)) was performed with the TaqMan™ allele-specific amplification method (Life Technologies, Carlsbad, CA), as previously described [32].” (Materials and methods: Genotyping)

#15. Line 112: were PCR protocols optimised or were they identical to reference 30? Gel % and stain type, primers all the same?

Response:

Reference 30 analyzed the same sample of our study. Therefore, the PCR protocols are identical to that of reference 30. (now updated as reference 32)

#16. Line 144: stats – was relatedness between individuals accounted for? i.e. if there was a mother and grandmother that were themselves mother and daughter. If participants were not related specify this in participant info section. If individuals were related, consider a MCMC-based Bayesian GLMM with priors to reduce chance of type 1 errors. If this stats test is not possible comment on how relatedness was accounted for in the analysis.

Response:

Thank you for pointing out the important statistical considerations and further suggestions. We took into account the sample clustering within families by estimating standard error using a robust variance estimator as stated in the statistical analysis section.

“To take into account clustering at the family level and age differences, the sandwich variance estimator was applied, and age was adjusted for all models.” (Materials and methods: Statistical analysis)

#17. Line 145: unclear why multivariate linear regression was used in addition to ANOVA when the variables used were categorical (genes and sex) and therefore more suited to the ANOVA. If these are not categorical perhaps a data prep section would help with clarification here.

Response:

First, essentially, we note that linear regression and ANOVA do the same mathematical calculations and have no superiority to each other. We mentioned our statistical procedure as multivariate linear regression simply because we run with the R function “lm”, which provides the same results as the R function “aov”, a function designed solely for ANOVA. Second, we preferred linear regression outputs here since this provided a difference score, which notified the readers of the strength of impact given the trust score range. This was followed up with a graphical description, giving the information about scores of each group.

#18. Line 151: reference needed for package

Response:

We added the citations where applicable.

“According to a power calculation with the “pwr” package in R [35],36 to 46 participants in each group were needed to detect a medium to large effect size (f2 = 0.12 to 0.16) with 80% power for an F-test with 4 groups, at the significant level set to 0.001 (= 0.05/ 8 gene combinations, 3 trust types, 2 sex; based on the Bonferroni correction).” (Materials and methods: Genotyping)

“ANOVA was followed by Tukey’s multiple comparison post hoc tests to identify groups with significantly different levels of trust with the R package “multcomp” [38].” (Materials and methods: Statistical analysis)

33. Champely S. pwr: Basic Functions for Power Analysis. 2020. Available: https://CRAN.R-project.org/package=pwr

36. Westfall THFB. Simultaneous Inference in General Parametric Models. Biometrical Journal. 2008;50: 3446–3363.

#19. Line 158-159: no test statistic or dfs reported with results.

Response:

I added the effect size and t-statistics in addition to the p-values in the result.

“However, the interaction between OXTR rs1042778, COMT rs4680, and sex on neighborhood trust (unstandardized coefficient (B) = 26.38, t = 2.40, p = 0.02), and between OXTR rs2254298, DRD2 rs1800497, and sex on institutional trust (B = -21.04, t = -2.54, p = 0.01) were found.” (Results: paragraph 1)

#20. Line 162: which test does the B value come from. As far as I am aware you get t from LR and f from ANOVA. If another test was conducted this needs reported in the stats section. If B is something other than the test statistic this needs explained here. Same in Tables 2 and 3.

Response:

B corresponds to the unstandardized effect size in the linear regression. I made a note for the first appearing B in the text and put a footnote to the table to explain what B stands for.

“However, the interaction between OXTR rs1042778, COMT rs4680, and sex on neighborhood trust (unstandardized coefficient (B) = 26.38, t = 2.40, p = 0.02), and between OXTR rs2254298, DRD2 rs1800497, and sex on institutional trust (B = -21.04, t = -2.54, p = 0.01) were found.” (Results: paragraph 1)

#21. Line 162: Consider “…among women (B=1.42, p = 0.86; Table 2).” Rather than having brackets back to back. Same on line 166 and 172.

Response:

I checked throughout the result section and made changes as you suggested where applicable.

#22. Line 164-165: report df for genotypes and for M&F for the genotypes on line 169-172.

Response:

I showed t-statistics and p-values in the results section. P-values are derived from t-statistics and the degree of freedom; therefore we avoided showing the degree of freedom here since it is redundant. 

#23. Line 190 (OXTR discussion): consider also OXTR has been shown to be linked with social attention in rhesus macaques (Howarth et al., 2023;10.1371/journal.pone.0288108) suggesting that this is a well conserved social response across species.

Response:

Thank you for showing the important literature. I am aware of the importance of being supported in animal experiments and included the argument regarding the suggested literature in the discussion.

“The association between OXTR and social attention was also shown in an animal experiment [44], suggesting this as a well-conserved social response across species.” (Discussion: paragraph 2)

#24. Line 219: A allele carries for OXTR rs53576 were also had lower level of optimism and self-esteem (Saphire-Bernstein et al., 2011; https://doi.org/10.1073/pnas.1113137108) and have poor social recognition (Skuse et al., 2014; 10.1073/pnas.1302985111).

Response:

Thank you for showing the important literature to be included in the manuscript. We added this research.

“For example, in one study OXTR rs53576 A allele carriers showed more social connectedness [14] while in another study A allele carriers exhibited lower empathy [15] and lower levels o

---

## [Decision Letter · Decision Letter 1]

30 Jul 2024

Epistatic interactions between oxytocin- and dopamine-related genes and trust

PONE-D-23-41164R1

Dear Dr. Fujiwara,

We’re pleased to inform you that your manuscript has been judged scientifically suitable for publication and will be formally accepted for publication once it meets all outstanding technical requirements.

Kind regards,

Alexandra Kavushansky, PhD

Academic Editor

PLOS ONE

Additional Editor Comments (optional):

Reviewers' comments:

Reviewer's Responses to Questions

**Comments to the Author**

1. If the authors have adequately addressed your comments raised in a previous round of review and you feel that this manuscript is now acceptable for publication, you may indicate that here to bypass the “Comments to the Author” section, enter your conflict of interest statement in the “Confidential to Editor” section, and submit your "Accept" recommendation.

Reviewer #2: All comments have been addressed

2. Is the manuscript technically sound, and do the data support the conclusions?

Reviewer #2: Yes

3. Has the statistical analysis been performed appropriately and rigorously? 

Reviewer #2: Yes

4. Have the authors made all data underlying the findings in their manuscript fully available?

Reviewer #2: Yes

5. Is the manuscript presented in an intelligible fashion and written in standard English?

Reviewer #2: Yes

6. Review Comments to the Author

Reviewer #2: I am happy that all of my comments have been addressed and this interesting paper is now ready for publication. My best to the authors.

7. PLOS authors have the option to publish the peer review history of their article (what does this mean?). If published, this will include your full peer review and any attached files.

Reviewer #2: **Yes: **Emmeline Howarth

---

## [Editor Report · Acceptance letter]

5 Aug 2024

PONE-D-23-41164R1 

PLOS ONE

Dear Dr. Fujiwara, 

I'm pleased to inform you that your manuscript has been deemed suitable for publication in PLOS ONE. Congratulations! Your manuscript is now being handed over to our production team.

Kind regards, 

on behalf of

Dr. Alexandra Kavushansky 

Academic Editor

PLOS ONE